# Cell Heterogeneity Analysis Revealed the Key Role of Fibroblasts in the Magnum Regression of Ducks

**DOI:** 10.3390/ani14071072

**Published:** 2024-04-01

**Authors:** Xue Du, Xiaoqin Xu, Yali Liu, Zhijun Wang, Hao Qiu, Ayong Zhao, Lizhi Lu

**Affiliations:** 1Key Laboratory of Applied Technology on Green-Eco-Healthy Animal Husbandry of Zhejiang Province, Zhejiang Provincial Engineering Laboratory for Animal Health Inspection & Internet Technology, Zhejiang International Science and Technology Cooperation Base for Veterinary Medicine and Health Management, China-Australia Joint Laboratory for Animal Health Big Data Analytics, College of Animal Science and Technology & College of Veterinary Medicine of Zhejiang A&F University, Hangzhou 311300, China; duxue@zafu.edu.cn (X.D.);; 2Institute of Ecology, China West Normal University, Nanchong 637002, China; 3Zhejiang Provincial Animal Husbandry Technology Promotion and Breeding Livestock and Poultry Monitoring Station, Hangzhou 310020, China; 4Independent Researcher, Hangzhou 310021, China; 5Key Laboratory of Livestock and Poultry Resources (Poultry) Evaluation and Utilization, Ministry of Agriculture and Rural Affairs of China, State Key Laboratory for Managing Biotic and Chemical Threats to the Quality and Safety of Agro-Products, Institute of Animal Science & Veterinary, Zhejiang Academy of Agricultural Sciences, Hangzhou 310021, China

**Keywords:** ceased-laying duck, oviduct regression, *THY1*, *TIMP4*, cell heterogeneity

## Abstract

**Simple Summary:**

This study investigated the molecular mechanisms of oviduct regression in ducks. Single-cell transcriptome sequencing of magnum tissue from egg-laying and ceased-laying ducks revealed significant heterogeneity, particularly in protein secretion cells and ECM-producing fibroblasts. Egg-laying ducks exhibited more protein secretion cells, crucial for albumen deposition, and higher proportions of *THY1*^+^ and *TIMP4*^+^ fibroblasts compared to ceased-laying ducks. These findings imply a correlation between *THY1* and *TIMP4* expression in fibroblasts and oviduct activity during reproduction. The study provides valuable insights into reproductive tract degeneration and potential improvements in laying duck production efficiency.

**Abstract:**

Duck egg production, like that of laying hens, follows a typical low–peak–low cycle, reflecting the dynamics of the reproductive system. Post-peak, some ducks undergo a cessation of egg laying, indicative of a regression process in the oviduct. Notably, the magnum, being the longest segment of the oviduct, plays a crucial role in protein secretion. Despite its significance, few studies have investigated the molecular mechanisms underlying oviduct regression in ducks that have ceased laying eggs. In this study, we conducted single-cell transcriptome sequencing on the magnum tissue of *Shaoxing ducks* at 467 days of age, utilizing the 10× Genomics platform. This approach allowed us to generate a detailed magnum transcriptome map of both egg-laying and ceased-laying ducks. We collected transcriptome data from 13,708 individual cells, which were then subjected to computational analysis, resulting in the identification of 27 distinct cell clusters. Marker genes were subsequently employed to categorize these clusters into specific cell types. Our analysis revealed notable heterogeneity in magnum cells between the egg-laying and ceased-laying ducks, primarily characterized by variations in cells involved in protein secretion and extracellular matrix (ECM)-producing fibroblasts. Specifically, cells engaged in protein secretion were predominantly observed in the egg-laying ducks, indicative of their role in functional albumen deposition within the magnum, a phenomenon not observed in the ceased-laying ducks. Moreover, the proportion of *THY1*^+^ cells within the ECM-producing fibroblasts was found to be significantly higher in the egg-laying ducks (59%) compared to the ceased-laying ducks (24%). Similarly, *TIMP4*^+^ fibroblasts constituted a greater proportion of the ECM-producing fibroblasts in the egg-laying ducks (83%) compared to the ceased-laying ducks (58%). These findings suggest a potential correlation between the expression of *THY1* and *TIMP4* in ECM-producing fibroblasts and oviduct activity during functional reproduction. Our study provides valuable single-cell insights that warrant further investigation into the biological implications of fibroblast subsets in the degeneration of the reproductive tract. Moreover, these insights hold promise for enhancing the production efficiency of laying ducks.

## 1. Introduction

The avian oviduct, serving as the egg formation tract, exhibits distinct characteristics compared to its mammalian counterpart. Structurally, the poultry oviduct is divided into four functional regions: the infundibulum, magnum, isthmus, shell gland, and vagina, each with clearly defined roles in the egg formation process [1]. As laying ducks enter the late phase of egg production, there is a gradual cessation of egg-laying activity. Notably, the histomorphology of the poultry oviduct during the ceased-laying phase undergoes significant regression compared to the active egg-laying phase. 

To enhance laying duck production, a comprehensive understanding of the regulatory mechanisms governing the oviduct during egg formation is imperative. The cessation of egg laying is accompanied by diminished pituitary sensitivity to luteinizing hormone-releasing hormone (LHRH), and reduced plasma concentrations of luteinizing hormone (LH), estradiol, and progesterone [2]. Persistent disruptions to these biological controls precipitate ovarian and oviductal regression, resulting in the permanent cessation of egg production. Disruptions of elemental homeostasis lead to the accumulation of DNA damage, ultimately resulting in organ dysfunction [3]. Over the course of the development of the chicken oviduct, B-cell lymphoma 2, proliferating cell nuclear antigen, survivin-142, and some caspases cooperatively orchestrate the cell proliferation, apoptosis, and differentiation [4]. Oviductal regression is a complex process involving the remodeling and degradation of the extracellular matrix (ECM), with ECM-receptor interactions playing a crucial role in the development of asymmetric poultry ovaries [5]. The extracellular matrix (ECM), mainly produced by fibroblasts, is a group of extracellular macromolecules composed of collagen, fibronectin, laminin, elastin, and other proteins. It provides structural and biochemical support to cells, regulating the phenotype and function of various cell types [6,7]. The matrix metalloproteinase (MMP) system is indispensable for ECM degradation during tissue remodeling. Studies on chicken oviduct regression have reported elevated relative mRNA levels of MMP-2, MMP-7, and MMP-9, as well as tissue inhibitors of metalloproteinases (TIMP-2 and TIMP-3) [2]. 

Accompanied by histomorphological changes, the oviduct undergoes significant functional alterations. The magnum, situated between the infundibulum and isthmus, plays a crucial role in protein synthesis and secretion for albumen formation in laying ducks. Yin et al. [8] identified five genes as the most highly expressed in the magnum of laying chickens: TF (ovo-transferrin precursor), OVAL (ovalbumin), LYZ (lysozyme C), SPINK7 (ovomucoid), and ORM1 (orosomucoid 1). In the aging process of laying hens, the decrease in the utilization of selenium and manganese affected amino acid metabolism, and accelerated magnum cell senescence by triggering necroptosis activation, thereby diminishing albumen secretion from the magnum [9]. Furthermore, the magnum holds tremendous value and potential as a bioreactor for exogenous protein production [10].

The majority of functional genes were previously discovered through studies on bulk tissue or cultured cells. However, investigating individual cell functional genes is crucial for elucidating the mechanism transformation of function. High-throughput scRNA-seq (single-cell RNA sequencing) allows for the examination of cell heterogeneity that is obscured in population-averaged measurements. Recently, scRNA-seq has been increasingly applied in research on genetic evolution, disease mechanisms, growth, and development. In the field of animal sciences, scRNA-seq has been utilized to characterize the heterogeneity of skeletal muscle at different developmental stages of chicken [11]. Previously, we reported the heterogeneity in duck liver at different egg-laying periods, with hepatocytes exhibiting dominant heterogeneity among all cell types [12]. In the present study, we constructed a single-cell transcriptome map of the duck magnum, revealing cell heterogeneity between egg-laying and ceased-laying ducks, and identified a potential correlation between the expression of THY1 (Thymus Cell Antigen 1, CD90) and TIMP4 (Tissue Inhibitor Of Metalloproteinases 4) in ECM-producing fibroblasts and oviduct activity during functional reproduction. The results provide valuable single-cell insights into fibroblast subsets in the degeneration of oviduct, which will hold promise for enhancing the production efficiency of laying ducks.

## 2. Materials and Methods

### 2.1. Animals

At 450 days old, 50 female individuals from one hereditary line of *Shaoxing ducks* were randomly selected as described in our previous study [12]. These ducks were housed individually in cages under natural temperature and light conditions, with ad libitum access to food and water. Egg production was monitored continuously for a period of 12 days. During this timeframe, some ducks ceased laying eggs. At 462 days old, one laying duck with fully functional oviducts and one ceased-laying duck with regressed oviducts (having ceased laying for a minimum of 7 days) were randomly chosen for sacrifice by cervical dislocation to obtain magnum samples. The laying duck was designated as O_L, while the ceased-laying duck was designated as O_C. The oviduct morphologies of both the egg-laying and ceased-laying ducks are provided in Appendix A. All the experimental procedures related to animals employed in this work strictly followed the Ethics Committee for Animal Experiments of Zhejiang A&F University, and were carried out in compliance with their Guidelines for Animal Experimentation. 

### 2.2. Cell Preparation

One square centimeter of magnum from each duck was dissected and rinsed with cold PBS to remove the mucus. The tissues were divided into small pieces and incubated with 0.25% trypsin–EDTA (Cat.No.GNM2520, GENOM BIO, Shanghai, China) on ice for 10 min. Next, they were gently agitated for 90 min at 37 °C while being digested with 0.5% Collagenase Ⅰ (Cat.No.17100019, Invitrogen, Waltham, MA, USA) after being rinsed with cold FBS (Cat.No.10099141C, Gibco, Melbourne, Australia). A 40 μm cell strainer was used to filter the suspensions to create the single-cell suspensions. Then, the single-cell suspensions were centrifuged at 4 °C for 10 min in 300× *g* centrifugation, before being treated with 1× RBC lysis buffer for 10 min to obtain the sedimentary cells. Single cells were cleaned before being suspended at 7 × 10^5^ cells/cm^2^ in cold DPBS (Cat.No.GNM14190, GENOM BIO, China), with 1% BSA (Cat.No.PC0001, Solarbio, Beijing, China). 

### 2.3. Library Construction

The sing-cell suspensions were loaded onto the Single Cell Controller (10× Genomics Chromium Single Cell System, 10× Genomics, San Francisco, CA, USA) to generate single-cell Gel Beads-In-Emulsions (GEMs). cDNA generation, amplification, and sequence break were carried out following the instructions of the Chromium Single Cell 3’v2 Reagent Kit (Cat# 1000121, 10× Genomics, USA). Indexes were added using the reagents of the GemCode Single-Cell 3′ Library Kit. After being quantified by quantitative PCR, sequencing libraries were sequenced by an Illumina NextSeq 500 with paired-end kits. Reads 1 in each sequence demonstrated the different transcripts of different cells. Reads 2 determined the genetic information. See our previous study for details [12].

### 2.4. Quality Assessment and Data Processing

The Q30 value was detected. The Cell Ranger software v2 (10× Genomics) was used to perform de-multiplexing of the sequences, adaptor trimming, and mapping to obtain the gene/cell count matrices. The STAR [13] helped to align the reads with the reference genome (Anas platyrhynchos, https://www.ncbi.nlm.nih.gov/assembly/GCF_003850225.1/ (accessed on 3 April 2021)). The Maximal Mappable Prefix from the first base of the read was determined [14]. The number of reads that provided meaningful information was calculated. Analyses were performed using the Seurat software v2 with default parameters [15]. The cells were filtered by CellRanger within a threshold (gene number > 200; UMI number < 30,000; mitochondrial expression < 0.5.)

### 2.5. PCA and t-SNE Demonstration

Dimensionality reduction, cell clustering, and differential gene expression analysis were performed using Seurat software version 2. We performed principal component analysis (PCA) on the genes sorted by their normalized dispersion to cluster and visualize the data, following a methodology similar to that of Macosko et al. [16]. Using the Uniform Manifold Approximation and Projection (UMAP) algorithm, a two-dimensional map of the cell populations was generated [17]. Cells were clustered using the Seurat package’s FindClusters function, with a resolution parameter set to 0.6 for both groups. The Seurat FindMarkers function was used to find the differential genes of each cluster, and the Wilcoxon rank sum test was used to calculate the significant level. Genes with |Log2FC > 0.25| and adjusted *p*-values less than 0.01 were considered differentially expressed genes. A heatmap and violin plot were constructed using the top 20 up-regulated genes in each cluster. The magnum is unique to birds, so there are no reference data in the databases that are currently in use. In order to correlate known cell types with cluster specific genes, we read as many papers as we could.

## 3. Results

### 3.1. Sequence Quality Control and Statistics 

The libraries produced a total of 72.8 million reads, indicating high quality, with valid barcodes exceeding 93%, Q30 bases in the barcode over 95%, Q30 bases in the RNA read exceeding 93%, and Q30 bases in the unique molecular identifiers (UMI) surpassing 94% (Appendix A). On average, 1309 and 964 genes were identified per cell in O_C and O_L, respectively, with over 71.7% of reads overall matching the reference genome. Appendix A illustrates the median number of genes detected from each cell plotted against the mean reads from each cell. O_C exhibited an average of 49,347 reads per cell with a total of 19,490 genes identified, while O_L showed an average of 58,094 reads per cell with 18,270 total genes detected (Appendix A).

### 3.2. Cell Clustering

The Cell Ranger software v2 was used for cell-filtering-based predefined thresholds, including the gene number (>200), UMI number (<30,000), and mitochondrial expression (<0.5). Following quality screening, a total of 13,432 single cells were retained, comprising 7319 cells from O_C and 5113 from O_L (Appendix A). Principal components analysis identified 10 primary factors, and subsequent clustering using Uniform Manifold Approximation and Projection (UMAP) categorized cells into 27 clusters (Figure 1A and Appendix A). The distribution of cell clusters was found to be greater in O_C compared to O_L (Figure 1A–C), indicating a higher level of cellular homogeneity within the tissue microenvironment of the magnum in the egg-laying ducks. Notably, certain clusters (4, 5, 7, and 11) were exclusively observed in O_L, while others (9, 15, 19, 22, and 25) were specific to O_C. Furthermore, a heatmap was generated to visualize the normalized expression of the top two differentially expressed genes across the 27 clusters (Figure 1D), revealing distinct patterns of gene expression for each cluster. 

### 3.3. Differential Analysis of Cells between Two Groups

Pearson correlation analysis of the gene expression levels among the different cell clusters revealed the highest association coefficients for clusters 5, 7, and 11 (r > 0.99), with clusters 6 and 23 exhibiting moderately high associations (r = 0.97) (Figure 2A). Clusters 5, 7, and 11, characterized by high expression levels of the *TF* and *LYZ* genes, are potentially involved in the secretion of transferrin and lysozymes (Figure 2B,F). Cluster 4 represents a cell cluster with high expression of the *MUC6* gene, suggesting its potential role in mucin secretion (Figure 2B). Notably, only the egg-laying ducks exhibited high expression of the *MUC6*, *TF*, and *LYZ* genes, while the ceased-laying ducks lacked these characteristics (Figure 1A and Figure 2B), aligning with the role of albumen deposition in the magnum of egg-laying ducks. Both Cluster 6 and Cluster 23 are identified as fibroblasts based on their high expression of marker genes. Cluster 6 exhibited high expression of *THBS1* (gene of thrombospondin), *DCN* (gene of decorin), *COL1A2* (gene of collagen type I alpha 2 chain), *FRZB* (gene of frizzled-related protein), *COL3A1* (gene of collagen type III alpha 1 chain), *MGP* (gene of matrix Gla protein), *GSN* (gene of gelsolin), and *TNXB* (gene of tenascin XB) (Figure 2C), whereas Cluster 23 shows high expression of *APOA1* (gene of apolipoprotein A1), *POSTN* (gene of periostin), *THY1* (gene of thymic cell differentiation antigen, THY1/CD90), *HPGDS* (gene of hematopoietic prostaglandin D synthase), *PODXL* (podocalyxin-like gene), *RRAD* (gene of Ras-related glycolysis inhibitor and calcium channel regulator), and *CLDN2* (gene of claudin 2) (Figure 2D). Cluster 6 comprises 877 fibroblasts, while Cluster 23 comprises 159 fibroblasts (Appendix A). Interestingly, in both O_C and O_L, more than 90% of the fibroblasts from Cluster 23 express the *THY1* gene (Figure 2F-up). However, in Cluster 6, *THY1*^+^ fibroblasts are predominant in the magnum of the egg-laying ducks (59%), while *THY1*^-^ fibroblasts dominate in the magnum of the ceased-laying ducks (76%) (Figure 2F-down). 

From the single-cell transcriptional data, we detected the expression of eight matrix metalloproteinases (MMPs) genes and two tissue inhibitors of metalloproteinases (TIMPs) genes, specifically *MMP2*, *MMP9*, *MMP11*, *MMP15*, *MMP16*, *MMP17*, *MMP23B*, *MMP28*, *TIMP3*, and *TIMP4*. Among these, Cluster 6 exhibited expression of *MMP2*, *TIMP3*, and *TIMP4*. The proportions of cells expressing *MMP2*, *TIMP3*, or *TIMP4* in O_L and O_C are depicted in Figure 3. Of particular interest is the observation that the proportion of *TIMP4*^+^ fibroblasts in Cluster 6 of O_L is 17%, while in Cluster 6 of O_C, it is 42%. 

Cluster 27 was identified as endothelial cells, characterized by the expression of RAMP2 (receptor activity modifying protein 2) [18]. Clusters 3, 16, 18, 25, and 26 were classified as immune cells, exhibiting expression of *SRGN*^+^ (serglycin) [19], *CXCR4*^+^ (C-X-C motif chemokine receptor 4) [19], and *WIPF*^+^ (WAS/WASL interacting protein family member 1) [19]. Clusters 8, 13, and 14 were identified as ciliated cells, expressing *SYNE1*^+^ (spectrin repeat containing nuclear envelope protein 1), *SNTN*^+^ (sentan, cilia apical structure protein), *CFAP46*^+^ (cilia and flagella associated protein 46), and *CROCC*^+^ (ciliary rootlet coiled-coil, rootletin) [20]. 

## 4. Discussion

In this study, we performed scRNA-seq on 13,708 individual cells extracted from the magnum tissue of egg-laying and ceased-laying *Shaoxing Ducks* using 10× Genomics technology. Through statistical analysis, we identified genes that were differentially expressed among these cells. Subsequently, we categorized the cells into distinct clusters based on their gene expression profiles, enabling us to analyze the cellular composition and status of the magnum tissue between the egg-laying and ceased-laying ducks at the single-cell level. The transcriptional profiles of fibroblasts, ciliated cells, endothelial cells, immune cells, *MUC* high cells, and *LYZ* and *TF* high cells from the magnum were examined within the same batch. As the tissue of magnum is specific to birds, existing databases lack reference information on bird single cells. To infer cell types, we extensively reviewed the relevant literature. Clusters 3, 16, 18, 26, and 27 are likely involved in basic functional maintenance and are present in both the egg-laying and ceased-laying ducks. Our comparative analysis of single-cell transcriptomes of the magnum of egg-laying and ceased-laying ducks revealed heterogeneous expression patterns of the *THY1* and *TIMP4* in ECM-producing fibroblasts. 

The magnum serves two primary functions: the secretion of albumen and propulsion of the egg forward. Our findings reveal distinct disparities in cell types and functionalities between the magnum of egg-laying and ceased-laying ducks. Specifically, notable differences are observed in the heterogeneity of protein-secreting cells and fibroblast cells. The role of protein secretion has been widely studied and partially applied previously [21,22]. So, the function of fibroblasts has been thoroughly analyzed in this study.

Fibroblasts are integral mesenchymal cells primarily responsible for synthesizing and remodeling the extracellular matrix (ECM), which significantly influences tissue morphology and function [7]. Studies on human scar tissue by Liu J et al. [23] have indicated that *CTHRC1*^+^ fibroblasts may exacerbate scarring by excessively depositing ECM under the influence of macrophage stimulation. Furthermore, research has underscored the critical role of fibroblasts in facilitating the growth, differentiation, and repair of functional cells such as epithelial and smooth muscle cells [24]. Our study identified two distinct clusters of fibroblasts. Cluster 6 exhibited high expression of *THBS1*, *DCN* [25], *FRZB*, *COL1A2*, *COL3A1*, *MGP*, *GSN*, and *TNXB*. These fibroblasts displayed heightened levels of collagen type I alpha 2 chain, thrombospondin 1, decorin, frizzled-related protein, collagen type III alpha 1 chain, matrix Gla protein, gelsolin, and tenascin XB, all of which are pivotal components of the ECM [26]. Cluster 23, on the other hand, demonstrated elevated expression of *POSTN* [27] and *CLDN2*, contributing to the production of periostin and claudin 2, which play significant roles in cell adhesion. Notably, we observed a disparity in the proportion of *THY1*^+^ fibroblasts within ECM-producing fibroblasts (Cluster 6) between the egg-laying ducks and ceased-laying ducks. In the egg-laying ducks, the ratio of *THY1*^+^ cells in the ECM-producing fibroblasts was 59%, more than double the ratio observed in the ceased-laying ducks (24%). THY1 is known to regulate the development and differentiation of various cells, as well as intercellular and cell-matrix adhesion and deformation [28,29]. *THY1*^+^ fibroblasts exhibit high heterogeneity and are implicated in physiological and biochemical processes such as tumor progression, fibrosis, and cellular proliferation [30]. Aoshima et al. [31] found that the expression levels of ECM-related genes (*COL1A1*, *COL3A1*, *FN1*, and *ELN*) in mouse *THY1*^+^ myofibroblasts were significantly higher than those in THY1^-^ myofibroblasts. *THY1*^+^ fibroblasts can produce more latent TGF-β and selectively activate it, thereby inducing ECM remodeling [32]. On the other hand, THY1 affects the secretion of ECM by interacting with receptors such as integrins and polysaccharide 4 [33]. Thus, the decline in the proportion of *THY1*^+^ fibroblasts following magnum regression suggests a potential association between the ratio of *THY1*^+^ fibroblasts and the functional reproductive activities of the oviduct. 

On the other hand, MMPs and TIMPs play pivotal roles in ECM degradation during tissue remodeling. In our study, eight *MMP*s and two *TIMP*s were detected. The ECM-producing fibroblasts expressed *MMP2*, *TIMP3*, and *TIMP4*, with only *TIMP4* showing differential expression between the egg-laying and ceased-laying ducks. Notably, in the ceased-laying ducks, the proportion of *TIMP4*^+^ cells within the ECM-producing fibroblasts (42%) was more than double that observed in the egg-laying ducks (17%). Previous investigations into ECM remodeling have demonstrated the significance of *TIMP4*. The deletion of *TIMP4* in female mice resulted in altered ECM composition, with notably elevated levels of COL1A1 and COL3A1 proteins compared with wild types [34]. Our findings further support the potential involvement of ECM-producing fibroblasts in the functional reproductive activities of the oviduct. 

Regrettably, we did not detect the muscle cells; this might have been caused by limited cell digestion. Moreover, some of the cell clusters were unclear. Liu et al. [35] reviewed the innovation of single-cell sequencing and its application in poultry science. We concur with their viewpoint that the limited availability of platforms for single-cell data in the poultry industry poses a barrier to their efficient utilization.

## 5. Conclusions

In summary, this study represents the first comprehensive profiling of cell heterogeneity within the magnum of ducks. The higher ratio of *THY1*^+^ cells/*TIMP4*^-^ cells within ECM-producing fibroblasts in egg-laying ducks compared to ceased-laying ducks suggests their potential involvement in the functional reproductive activities of the oviduct. These findings provide valuable single-cell insights that lay the groundwork for further exploration of the biological significance of fibroblast subsets in magnum regression in ducks. Moreover, this research has the potential to inform and guide the production of laying ducks.

## Figures and Tables

**Figure 1 animals-14-01072-f001:**
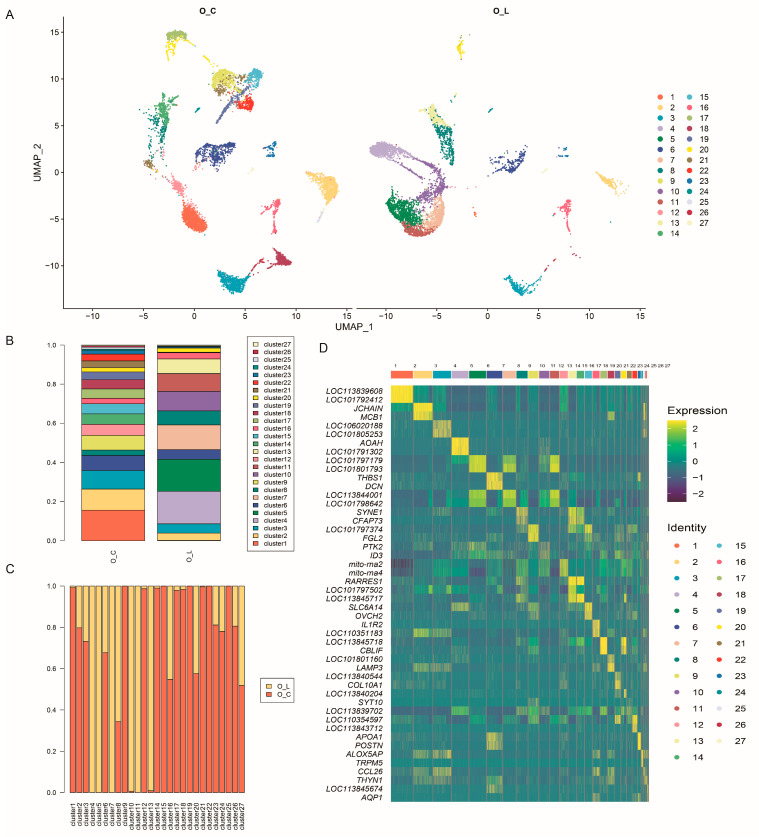
Comparative analysis of magnum cells between egg-laying and ceased-laying ducks. (**A**) UMAP analysis of duck magnum cell data. Cells are colored by cell clusters. The left image shows the cell clusters of O_C, the right image shows the cell clusters of O_L. (**B**) The percentage of different cell clusters in two samples. Different colors represent distinct cell clusters, with the horizontal axis representing different groups, and the vertical axis indicating the proportion of cells. (**C**) The percentage of different cell clusters in two samples. Different colors represent different groups, with the horizontal axis representing different cell clusters, and the vertical axis indicating the proportion of cells. (**D**) A gene expression heatmap showing the expression of the top two differentially expressed genes. Columns correspond to individual cell clusters, and rows represent the top 2 genes in each cluster. The colors ranging from blue to yellow represent an increase in expression level. O_C, magnum of ceased-laying duck. O_L, magnum of egg-laying duck.

**Figure 2 animals-14-01072-f002:**
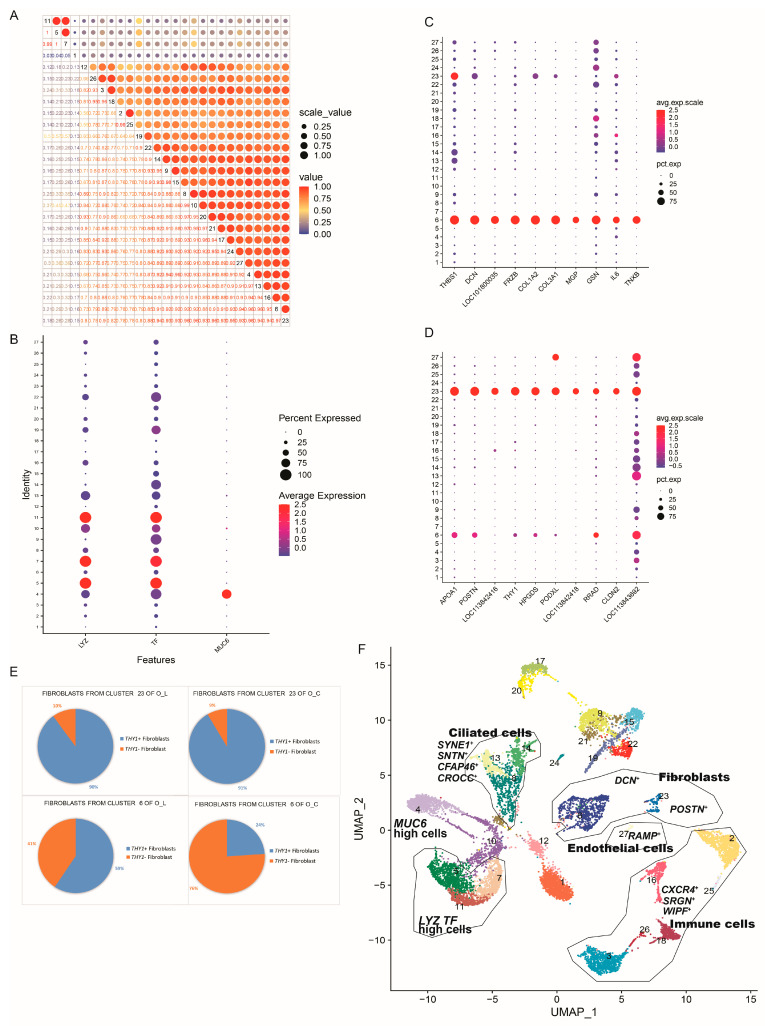
Details of specific clusters. (**A**) Correlation between cell clusters. The horizontal axis and vertical axis represent cell clusters, cluster name tags on the diagonal cells; (**B**) MUC6, TF, and LYZ displayed in different clusters; (**C**) Marker genes of Cluster 6 displayed in different clusters; (**D**) Marker genes of Cluster 23 displayed in different clusters; (**E**) Proportion of *THY1*^+^ fibroblasts and *THY1*^-^ fibroblasts in Clusters 6 and 23 of different groups; (**F**) UMAP shows cell types and their marker genes of different cell clusters. The numbers and the different colors represent the cell types. O_C, magnum of ceased-laying duck. O_L, magnum of laying-duck.

**Figure 3 animals-14-01072-f003:**
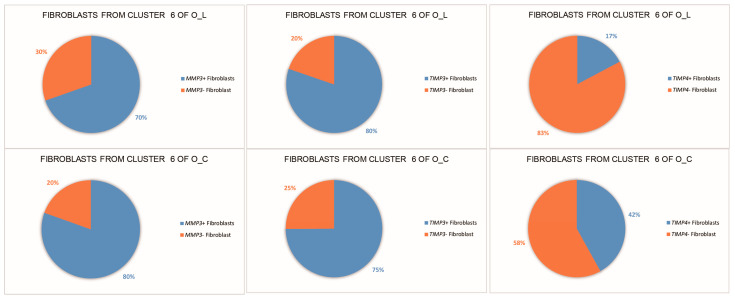
Proportion of *MMP2*^+^, *TIMP3*^+^ and *TIMP4*^+^ fibroblasts in Cluster 6 of different groups.

## Data Availability

The data that support the findings of this study are openly available in the NCBI Sequence Read Archive repository; the accession numbers are SRR25208343 and SRR25208342.

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
