# Peer review of "Cell Heterogeneity Analysis Revealed the Key Role of Fibroblasts in the Magnum Regression of Ducks"

_animals, 2024, doi:10.3390/ani14071072_

Round 1

Reviewer 1 Report

Comments and Suggestions for Authors

Research related to the study of Cell Heterogeneity analysis in ducks is a current area of research. The authors have done a lot of work. The reseach was completed at a high methodological level, but there are some questions:

1. The introduction does not clearly outline the purpose of the study, although the abstract presents it more clearly.

2. The methods do not indicate the number of samples studied

3. The correlation graph is not very informative (Fig2 A). Everything merges into one blue color. Authors are recommended to use a different color palette or present the graph in a other format.

Author Response

  1. The introduction does not clearly outline the purpose of the study, although the abstract presents it more clearly.

R 1. Thank you for pointing this out. Agree. The purpose of the study has been added in the Introduction part in revised version.

  1.  The methods do not indicate the number of samples studied

R 2. Thank you for pointing this out. Agree. One laing duck and one ceased-laying duck were sampled for the magnum study, as we described in the revised version: "At 462-day-old, one laying duck with fully functional oviducts and one ceased-laying duck with regressed oviducts (having ceased laying for a minimum of 7 days) were randomly chosen for sacrifice by cervical dislocation to obtain magnum samples. Laying duck was designated as O_L, while ceased-laying duck was designated as O_C."

  1. The correlation graph is not very informative (Fig2 A). Everything merges into one blue color. Authors are recommended to use a different color palette or present the graph in a other format.

R 3. Thank you for pointing this out. Agree. The color difference is more noticeable in the revised version.

Reviewer 2 Report

Comments and Suggestions for Authors

The mansucript is well written. Authors must explain in the experiemental design the house keeping genes and the control genes used and their relevance to the measurements conducted.

Some more recent references may be added in Intro and Discussion part.

Conclusions are well written. However, results may be driven to Discussion and Results part and conclusions to be usded for what authors find to cocnlude and future perspectives.

Author Response

  1. Authors must explain in the experiemental design the house keeping genes and the control genes used and their relevance to the measurements conducted.

R 1. Thank you for pointing this out. We don't fully understand this comments. However, no house keeping genes or control genes were used in the measurements.  As described in Material and methods  of last version "Each sequence contains Illumina P5 adaptor, P5 primer (TruSeq Read 1), Barcode (16 bp), UMI (12 bp), poly (dT) VN, cDNA, P7 primer (TruSeq Read 2), Index (i7 index read) and Illumina P7 adaptor. Reads 1 were used to distinguish different transcripts of different cells. Reads 2 were used to determine the genetic information.", we use the adaptors to tag the mRNAs. We obtained the expression of each gene in each cell through Data processing.

  1. Some more recent references may be added in Intro and Discussion part.

R 2. Thank you for pointing this out. Agree. We added references of last two years in Intro and Discussion part of revised version.

  1. Conclusions are well written. However, results may be driven to Discussion and Results part and conclusions to be usded for what authors find to cocnlude and future perspectives.

R 3. Thank you for pointing this out. Agree. We adjusted the exhibition of Results, Discussion and Conclusions.

Reviewer 3 Report

Comments and Suggestions for Authors

The authors have compared the cells of the magnum in laying and non-laying ducks.  I am broadly very impressed by the manuscript, the study and the innovative approach.  I have a series of concerns and stylistic issues that need to be addressed.

Concerns

1.     Figure 2 F UMAP showed cell types and their marker genes of 248 different cell clusters. O_C, magnum of ceased-laying duck. O_L, magnum of laying-duck. Data on O_C verses O_M are not clear.  Is something missing?

2.     Figure. 3 – I would encourage the authors to conduct statistical analysis to determine whether there are differences between magnum cells from ducks that were laying or had ceased to lay.

Minor and stylistic issues

1.     Line 76 to 77 “In aging laying hens, the reduced levels of selenium and manganese” citation needed.

2.     Lines 100-101 “At 450-day-old, female individuals from a hereditary line of Shaoxing ducks ”. citation needed.

3.     I question the authors use of the word “aging “ as cessation of egg laying may be viewed as part of photorefractoriness. 

4.     Line 94 “expression of THY1 and TIMP4 in ECM-producing fibroblasts and oviduct” Please define THY1 

5.     Lines 274-277. “In this study, we conducted single-cell RNA sequencing (scRNA-seq) on 13,708 individual cells extracted from the magnum tissue of egg-laying and ceased-laying Shaoxing Ducks using 10x Genomics technology. Through rigorous statistical analysis, we identified genes that were differentially expressed among these cells.” I would question the use of the word – “rigorous”. 

6.     Line 205 “blue corresponds to low expression level.”  There appear to be more than one shade of blue in the figure. 

7.     There are numerous case issues, for example the following: 

·      Line 77 “LYZ (Lysozyme C)” should be - LYZ (lysozyme C)

·      Line 343 “School research development Fund” should be - School Research Development Fund.

·      Line 425-426 “Advances in Single-Cell Sequencing Technology and Its Application in Poultry” should be - Advances in single-cell sequencing technology and its application in poultry.

8.     References

The authors should put the references into the format of the journal:

Journal names in references Journal Articles:
1. Author 1, A.B.; Author 2, C.D. Title of the article. Abbreviated Journal Name Year, Volume, page range.

 https://www.mdpi.com/journal/animals/instructions

Comments on the Quality of English Language

The authors have compared the cells of the magnum in laying and non-laying ducks.  I am broadly very impressed by the manuscript, the study and the innovative approach.  I have a series of concerns and stylistic issues that need to be addressed.

Concerns

1.     Figure 2 F UMAP showed cell types and their marker genes of 248 different cell clusters. O_C, magnum of ceased-laying duck. O_L, magnum of laying-duck. Data on O_C verses O_M are not clear.  Is something missing?

2.     Figure. 3 – I would encourage the authors to conduct statistical analysis to determine whether there are differences between magnum cells from ducks that were laying or had ceased to lay.

Minor and stylistic issues

1.     Line 76 to 77 “In aging laying hens, the reduced levels of selenium and manganese” citation needed.

2.     Lines 100-101 “At 450-day-old, female individuals from a hereditary line of Shaoxing ducks ”. citation needed.

3.     I question the authors use of the word “aging “ as cessation of egg laying may be viewed as part of photorefractoriness. 

4.     Line 94 “expression of THY1 and TIMP4 in ECM-producing fibroblasts and oviduct” Please define THY1 

5.     Lines 274-277. “In this study, we conducted single-cell RNA sequencing (scRNA-seq) on 13,708 individual cells extracted from the magnum tissue of egg-laying and ceased-laying Shaoxing Ducks using 10x Genomics technology. Through rigorous statistical analysis, we identified genes that were differentially expressed among these cells.” I would question the use of the word – “rigorous”. 

6.     Line 205 “blue corresponds to low expression level.”  There appear to be more than one shade of blue in the figure. 

7.     There are numerous case issues, for example the following: 

·      Line 77 “LYZ (Lysozyme C)” should be - LYZ (lysozyme C)

·      Line 343 “School research development Fund” should be - School Research Development Fund.

·      Line 425-426 “Advances in Single-Cell Sequencing Technology and Its Application in Poultry” should be - Advances in single-cell sequencing technology and its application in poultry.

8.     References

The authors should put the references into the format of the journal:

Journal names in references Journal Articles:
1. Author 1, A.B.; Author 2, C.D. Title of the article. Abbreviated Journal Name Year, Volume, page range.

 https://www.mdpi.com/journal/animals/instructions

Author Response

Concerns

  1. Figure 2 F UMAP showed cell types and their marker genes of 248 different cell clusters. O_C, magnum of ceased-laying duck. O_L, magnum of laying-duck. Data on O_C verses O_M are not clear.  Is something missing?

R 1. 1. Authors must explain in the experiemental design the house keeping genes and the control genes used and their relevance to the measurements conducted.

R 1. Thank you for pointing this out. Figure 2F UMAP showed all the cell types of the O_C and O_L and their marker genes. Data on O_C versese O_L are shown in Figure 1A-C. Specifically regarding the expression of a particular gene in different cell categories and groups, we only presented data for four genes: THY1, MMP2, TIMP3 and TIMP4, in Figure 2E and Figure 3 . 

  1. Figure. 3 – I would encourage the authors to conduct statistical analysis to determine whether there are differences between magnum cells from ducks that were laying or had ceased to lay.

R 2. We appreciate the reviewer's suggestion. Due to the limited sample size, with only one sample available, we are unable to conduct statistical analysis to determine whether there are significant differences between the protein-secreting cells and ECM-producing fibroblasts in ducks that are laying eggs compared to those that have ceased egg production. Our study primarily aims to demonstrate this trend and provide preliminary observational results. We will consider expanding the sample size for more in-depth statistical analysis in future studies.

 Minor and stylistic issues

  1. Line 76 to 77 “In aging laying hens, the reduced levels of selenium and manganese” citation needed.

R 1. This description is taken from reference [9], and we have cited it accordingly.

  1. Lines 100-101 “At 450-day-old, female individuals from a hereditary line of Shaoxingducks ”. citation needed.

R 3. Agree. One our previouse research has been cited in the revised version.

  1. I question the authors use of the word “aging “ as cessation of egg laying may be viewed as part of photorefractoriness. 

R 3. Agree. In the new version, we have toned down the emphasis on the concept of "aging".

  1. Line 94 “expression of THY1 and TIMP4 in ECM-producing fibroblasts and oviduct” Please define THY1 

R 4. Thank you for pointing this out. Agree. We defined THY1 and TIMP4 in the revised version.

  1. Lines 274-277. “In this study, we conducted single-cell RNA sequencing (scRNA-seq) on 13,708 individual cells extracted from the magnum tissue of egg-laying and ceased-laying Shaoxing Ducks using 10x Genomics technology. Through rigorous statistical analysis, we identified genes that were differentially expressed among these cells.” I would question the use of the word – “rigorous”. 

R 5. Thank you for pointing this out. Agree.  In the revised version, we have removed this word.

  1. Line 205 “blue corresponds to low expression level.”  There appear to be more than one shade of blue in the figure. 

R 6. Thank you for pointing this out. Agree. In the revised version, we have changed the way it is described.

  1. There are numerous case issues, for example the following: 
  • Line 77 “LYZ (Lysozyme C)” should be - LYZ (lysozyme C)
  • Line 343 “School research development Fund” should be - School Research Development Fund.
  • Line 425-426 “Advances in Single-Cell Sequencing Technology and Its Application in Poultry” should be - Advances in single-cell sequencing technology and its application in poultry.

 R 7. Agree. In the revised version, we have made the required modifications at the appropriate locations.

  1. References

The authors should put the references into the format of the journal:

Journal names in references Journal Articles:
1. Author 1, A.B.; Author 2, C.D. Title of the article. Abbreviated Journal Name YearVolume, page range.

 https://www.mdpi.com/journal/animals/instructions

R 8. Agree. In the revised version, we have made the required modifications in References.